# Multiculturalism and Women's Rights: Implications for Victims of Female Genital Cutting

**Mary Nyangweso**

Department of Philosophy and Religious Studies, East Carolina University, Greenville, NC 27858-4353, USA; wangilam@ecu.edu

**Abstract:** The evolution of the discourse surrounding human rights has led to calls for multiculturalism in modern society. While human rights originate from a perceived universal need to protect the rights of the individual, their appeal has not been universal, as they are perceived to be a threat to cultural rights by some. This is because of the perceived conflict or dilemma of negotiating both as entitlements. While arguments for both human rights and cultural rights are compelling, they expose a tension or conflict of rights. Calls for multiculturalism emerged in defense of cultural diversity and other forms of rights. The central question surrounding this tension is as follows: Can human and cultural rights be reconciled without compromising basic individual rights? Attempts to answer this question have occupied scholarship for several decades, with works on cross-cultural universalism and intersectionality emerging as a bridge for the seeming unbridgeable controversy. This essay explores some of these works that relate to the question of women's rights and the implications for the controversial practice of female genital cutting (FGC).

**Keywords:** women's rights; moral universalism; cultural relativism; female genital cutting

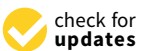



## 1. Introduction

As the scholarship on human rights has evolved, so have feminist positions. This evolution reflects the moral positions, namely, moral universalism and cultural relativism, that inform positions on human and cultural rights. These positions have evoked controversies due to the questions that are raised regarding how to achieve, promote, and protect human and cultural rights. As compelling as these positions are in presenting individual and cultural rights perspectives, they reveal the challenges and dilemmas implied by each position for women's rights. This dilemma is clearly articulated in the feminist discussions about female genital cutting (FGC).[1] As a socio-cultural practice, FGC, also known as female circumcision or female genital mutilation, is commonly found in some African, Middle Eastern, and Asian countries, where it involves the cutting or removal of part or all of the tissues around a woman's reproductive organ. It is performed on girls and young women as a rite of passage. While it is illegal in many of these countries, it has persisted and, in some cases, it has been introduced by migrants in European countries (Nyangweso 2014). Although it is condemned as a violation of human rights by those who embrace values associated with moral universalism due to the perceived health risks and sexist ideals it is associated with, it is embraced as a cultural right by those who embrace values associated with cultural relativism. The conflict in values that ensues highlights how significant and yet compelling the arguments for both positions are. Further, those who embrace the practice often find themselves in a dilemma regarding how to negotiate values and positions regarding moral universalism and cultural relativism values that shape both arguments (Nyangweso 2014). Some scholars argue for a cross-cultural universals approach as a way to bridge the seeming irreconcilable and unbridgeable positions. This essay examines the discourse on these positions and how they relate to women's rights and FGC.

## 2. Women's Rights Claim

Although the history of feminism as a movement is not very long, it has evolved and been shaped by several theoretical positions. Significant moral positions in this evolution include moral universalism and cultural relativism, which are theoretical frameworks that inform claims about human and cultural rights (Billet 2007). Moral universalism, also known as moral objectivism, refers to an ethical position that is considered to be universal and that its universal application transcends culture, race, sex, religion, nationality, sexual orientation, or any other distinguishing feature (Garth 2011). This position, which was promoted by philosophers such as Immanuel Kant (1993) and Richard Mervyn Hare ([1952] 1991), draws on moral imperatives that are regarded as equally binding on everyone (Kant 1993; Hare [1952] 1991). In Noam Chomsky's words: "If an action is right or wrong for others, it is right or wrong for us" (Chomsky 2002). As the ground and scope of moral judgment, moral universalism is the basis of universalistic ethics that informs human rights. As Jack Donnelly argued, moral universalism as principles set out certain standards of human behavior that are regularly protected as legal rights in national and international law (2007). To resist them on any basis is to overlook simple moral facts since all societies have cross-culturally and historically manifested conceptions of human rights in their contexts (Donnelly 2007, pp. 284–90).

To understand moral universalism, it is important to acknowledge the origins of universalism in the roots of Western liberalism and how this was informed by seismic shifts in Europe as a critique of communism and even how Christian values played a central role in this moral position. As argued by Larry Siedentop, the focus on the individual is a Western idea that gradually replaced claims to family, tribes, clans, and essentially group values. The belief in individual freedom and the fundamental moral equality of individuals and in legal systems based on equality and representative forms of governance is a Western idea that is sometimes found inadequate or irrelevant in some communities where group morality and values are embraced (Siedentop 2017). This background is necessary to understand the critiques that this position faces. Moral universalism as a theory of human rights is informed by four rational theories. The first theory, namely, the natural law theory, argues that human beings as equals possess a certain degree of sovereignty concerning ideals such as freedom and honor. Therefore, to impose any restriction on this sovereignty is morally wrong (Perry 1997, pp. 478–81; Billet 2007, p. 16). The second theory, namely, the theory of rationalism, argues that all humans, as rational beings, are equal to each other (Zechenter 1997, p. 321; Billet 2007, p. 1). The third, namely, the human capabilities theory, argues for the recognition of fundamental characteristics that define what it means to be human across diverse societies. As discussed by Martha C. Nussbaum, these characteristics include basic needs, such as food, drink, shelter, and mobility, and the capacity for pain and pleasure (1997). The fourth theory, the doctrine of positivism, argues that countries with representative forms of government should promote universal norms of behavior (Zechenter 1997, p. 321; Billet 2007, p. 6).

These four theories form the basis of universalistic ethics that has appealed to advocates of human rights and feminism. The perceptions of "sameness" or "everyone's alike" implied in this ethic soon encountered the challenge of traditionalism, privilege, and exclusion of some groups and women's experiences (Sevenhuijsen 1991). As Abdullah An Na'im explains, although the human rights doctrine that is central to moral universalism e has been influential within international law, it continues to provoke considerable skepticism and debates about the content, nature, and justifications of human rights to this day due to its origin and representation (An-Na'im 1996). Those who embrace moral universalistic values and human rights positions continue to grapple with criticisms and challenges that are based on the initial premises and assumptions. While the appeal to feminism was attractive, the application of this ethic was far from conforming to the values and principles of equality that feminism claimed to espouse.

This criticism influenced some feminists' position on the need for the establishment of non-sexist and non-discriminatory policies that were in line with the position of the United

Nations Declaration on the Discrimination against Women (CEDAW) (Steiner and Alston 1996, pp. 244–45). As C. Nagengast and T. Turner observed, CEDAW as an ideal continues to be a moral tool of cultural analysis for promoting equality of genders (Nagengast and Turner 1997, p. 270). However, as Annie Bunting argued, within the feminist scholarship, the tendency to dismiss cultural differences to maintain universalist and essentialist norms undermined the support for the feminist movement from the diverse international community worldwide. To ignore the importance of culture, race, class sexuality and history is to leave the feminist theoretically "impoverished and strategically weak" (Bunting 1993, p. 6).

Feminism, which has always been about women's rights claims, arose in condemnation of settled practices that undermined the rights and dignity of women. At question were the stunning levels of violence against women, ceaseless efforts to turn women's sexuality into a special burden, and disparities between economic opportunities. As a movement, it embraced moral universalism to advocate for women's rights. As Joshua Cohen et al. has observed, the basic feminist idea was to have women acknowledged as "human beings too". Women are to be treated as moral equals to men and their lives are not to be discounted as subordinate (Cohen et al. 1999). As the human rights movement has evolved and adapted to criticism that it has disregarded diversity, group rights, and multiculturalism, so has the feminist movement. This evolution has, over the past decade, seen the emergence of multiculturalism as the banner for condemning intolerance of other ways of life while encouraging diversity and the idea that people in other cultures, foreign and domestic, are human beings too. The advocate of moral equality sought the entitlement of all to equal respect and non-treatment of others as a subordinate caste.

Feminist scholars who are informed by the moral universalism ethic and human rights principles have characterized cultural practices such as FGC as a violation of women's rights. As explained by Ellen Gruenbaum, the practice is viewed as an expression of sexism and patriarchy. It is considered to be part of a social structure that legitimizes sexist cultural practices that include genital cutting, early and arranged child marriage, bride price, polygamy, and widow inheritance, which are all designed to limit women's self-realization and enjoyment of life (Gruenbaum 2001, p. 133). C. Coquery-Vidrovitch linked FGC to the notion of honor that is embraced in cutting communities, whose cultural values are intended to ensure that the children that are born by a woman are indeed her husband's and no one else's (Coquery-Vidrovitch 1997). FGC is particularly critiqued for devaluing female bodily pleasure to justify acts of sexual control. Thus, it is a practice that undermines individual rights to sexuality and reproductive rights. For instance, Alice Walker and P. Palmer described FGC as "the sexual blinding of women," to connote the destruction of female sexuality (Walker and Parmar 1993, pp. 15–19). Mary Daly considered the practice an "unspeakable atrocity, a torturous and mutilative practice that is, aimed at depriving women their femininity, sexual sensitivity, and pleasure" (Daly 1978, pp. 153–59). She characterized the removal of the clitoris and labia minora as the destruction of sensitive parts of the reproductive organ in some forms of FGC and as the interference with the right to sexual pleasure and reproductive health. Nancy Bonvillain described the procedure as sexually disfiguring because of the painful intercourse associated with the procedure and the disfiguring nature that the victim suffers when she loses part of her sexual organs in the process (Bonvillain 2001, p. 277). To Barbara S. Morrison, "the clitoris removal exposes the female body to unnecessary pain and it permanently obliterates one of the sites of pleasure that constitutes the female body", thus violating the right to sexual fulfillment (Morrison 2008, p. 126). While recognizing the unique African values, African feminism, which subscribes to a moral universalism stance, acknowledges social injustices that are embedded in some African cultural values and practices. They argue for claims of women's rights and the respect of individuals within these values systems. For instance, The African Charter on Human and Peoples' Rights (ACHPR) and the Maputo protocol, which drew upon the CEDAW, call for human rights with regard to transforming customary and traditional practices that violate women's rights. As per article 18(3) of the ACHPR charter, every nation is to ensure the elimination of every

discrimination against women and also to ensure the protection of the rights of the women and the child, as stipulated in international declarations and conventions. However, some African feminists have expressed concerns about the implications of liberal universalism for female genital cutting. As Musimbi Kanyoro and Rachel Angogo Kanyoro argued, African feminism has a hermeneutics that explains and analyzes the cultural experiences of African women in the African context (Kanyoro and Kanyoro 2002). This compelling argument is countered by multiculturalism, which argues for cultural rights. As Temelso Gashaw explained, efforts to outlaw discriminatory and harmful practices have also been equated with submission to neocolonial attitudes or failure to consider the cultural sensitivities of African society (Gashaws 2020). The fact that Article 17 of the same ACHPR charter seeks to protect and promote cultural and traditional values as recognized by a community invokes the cultural relativist argument. The African charter imposed an obligation on every individual to "preserve and strengthen positive cultural values" (Gashaws 2020). In seeking to accommodate universal rights with African values, the Charter attempts to reconcile universally recognized women's rights values (Gashaws 2020).

### 3. The Cultural Rights Argument

The cultural rights argument, also referred to as the group rights' position, is often used to promote cultural diversity or multiculturalism in defense of identity rights and anti-discrimination policies. Proponents of cultural diversity, such as Judith Butler, have argued for the need to recognize the reality of difference as a social reality that is embedded in cultural values that are expressed in a variety of social groups' experiences, such as gender, culture, race, religion, ethnicity, language, tradition, and morality (Butler 1990). They seek to redirect attention toward cultural values to guarantee social groups' access to their culture as a right. As argued by Iris Young, embracing difference is a condition for ensuring equality, human dignity, and no discrimination (Young 1992). The preservation of "traditional values" is often central to this position (Deveaux 2000, pp. 523–25).

As indicated earlier, the aim of multiculturalism is the protection of cultural diversity. Multiculturalism considers culture to be a strong part of people's lives because it shapes their values. The phrase "cultural diversity" emphasizes the quality of different cultures in a variety of human societies as opposed to monoculture. The failure of the minority groups to assimilate into majority cultures, as was widely expected, propagated multiculturalism as an approach that views the expectation of assimilation as oppressive. Multiculturalism is embedded in cultural relativism, a theory that views rights and rules about morality as encoded in and dependent on a particular socio-cultural context. As argued by Steiner and Alston, morality differs throughout the world because the different cultural values that inform them differ. Cultural relativism is a critique of cultures that seek to impose their ideals over others (Steiner and Alston 1996, pp. 192–93). This position arose in response to presumed biases against non-Western societies (An-Na'im 1996, pp. 210–20; Renteln 1990, pp. 62–63). Questions raised involve the basis for denying a cultural group the right to engage in its long-held cultural traditions. Scholars such as An-Na'im questioned the legitimacy of embracing human rights standards in cultures that not only consider them a violation of their fundamental beliefs but also about the way social life should be ordered. Some of these countries, notably African and Asian countries, did not participate in the formulation process that occurred in 1948 since they were victims of colonialism (An-Na'im 1996, p. 2011).

Kalev, Afkharmi, and Frida explains how cultural relativism as a discourse seeks to correct misconceived imperialist attitudes and assumptions that human rights are universally accepted (Kalev 2004, p. 345; Afkharmi and Friedl 1997). They explained how questions of difference, which characterized the 1994 Cairo Conference on Population and Development, represent the initial challenge to the concept of human rights. By invoking cultural imperialism and intellectual colonialism, scholars of cultural relativism questions the suitability of other cultures adopting human rights (Kalev 2004, p. 345; Afkharmi and Friedl 1997). As Holt explained, it is difficult to say that a certain kind of behavior is right or wrong for

all, as moral universalists claim, rather a certain kind of behavior is right or wrong relative to a specific society (2006). Influenced by this position, Gilbert and Ivancevich argued for inclusive policies that reflect fairness, respect, and commitment to the dignity of every person, regardless of their origin (Gilbert and Ivancevich 2002). Sebastian Poulter argues for the need to align policy with multicultural ideals to account for the cultural practices of minority communities and to "recognize and respect interests of liberal democracy" (Poulter 1986, p. 593).

Just like feminism was influenced by universalism, it was also influenced by cultural relativism. One position that emerges out of cultural relativism is intersectionality, which is a theory that advances transcendence over transfixed social boundaries and binary perceptions of reality to embrace social differences, such as gender, ethnicity, class, and sexual orientation. As a theoretical framework, intersectionality scholars, such as K. W. Crenshaw (1989, 1991) and Patricia Hill Collins (1990), argue for the need to recognize social ambiguity as an inherent social reality. For these scholars, an embrace of cultural diversity and multiculturalism is a legitimate approach since it endeavors toward the inclusion of all categories of social differentiation. Even as positions on moral universalism and human rights have evolved to ensure that the basic needs of the entire human population across all cultures are fulfilled, as Nussbaum argues, the question of what is "right" continues to ignite controversy, rendering the entire subject a philosophical debate (Nussbaum 1995, pp. 76–80). Value judgments made by outsiders in any culture are considered inappropriate, as argued by Ierodiaconou (1995). Despite the broadly shared characteristics of humanity and principles of justice that are attractive because of their insistence on the treatment of each person fairly, it is impossible to realize this fact across various cultural and regional boundaries, as argued by Alison Assiter (2016) and Martha Nussbaum (1995).

It is for this reason that cultural relativism was critiqued by scholars, such as Francis Beckwith and Gregory Koukl, who argued that it promotes a form of cultural romanticism that is resistant to cultural appraisal, which involves a critical reflection on cultural values to determine what to retain or discard (1998). An extreme form of cultural relativism can lead to a world in which nothing is to be considered to be wrong, they argues (1998). In such a world, justice and fairness become meaningless concepts since accountability is impossible (Beckwith and Koukl 1998, p. 69). John Tilley (2000) explains how some cultural relativism has promoted cultural determinism, which makes it difficult to compromise human and groups rights positions. In his opinion, cultural determinism is revised ethnocentricism, which renders "all of our beliefs, concepts and perceptions as culturally conditioned such that unbiased thoughts, choices, and inferences become impossible" (Tilley 2000, p. 540). Liberal political theorists, such as Kalev, argue that cultural relativism is likely to preserve the status quo, simply on the grounds of cultural norms because it fails to recognize that an individual can be oppressed as a member of a social group by norms that are acceptable in his or her community. To Kalev, individual members of a cultural community must be voluntary participants, entitled to leave their social groups if they so desire (Kalev 2004). Reiterating the same argument, Chandran Kukathas argues for the need for every individual to have the right to choose even those cultural practices that others may perceive as oppressive (Kukathas 1995, p. 234). To liberal political theorists, groups, cultural communities, or other such collectives must be perceived as mattering since their values and beliefs affect actual individuals (Kukathas 1995, p. 234). Ultimately, the fundamental question lies with the commitment to equality in a world of multiple human differences.

While the argument for social identity is compelling, the approach poses certain social implications for women's rights (Parekh 1997, p. 121; Grillo 1998, p. 189). As Iris Young argued, a social difference that is espoused by group rights is a source of power, a recognition, an identity, and an emancipation for the marginalized. It can also be a tool of oppression (Young 1992, pp. 39–63, 163). The tendency for multiculturalism to resist reconciliation efforts toward egalitarians' convictions undermines basic rights entitlement and calls for equal respect for all, especially in cultures that do not consider women as entitled to the same treatment as men. Some of these cultures believe that

unequal rights of ownership, political participation, vulnerability to violence, and access to educational opportunities are justified. This is what justifies Susan Okin's central question that relates to how feminism is to be sustained in a world that embraces multiculturalism. Is it possible to achieve a "multiculturalism that treats all persons like each other's moral equals?" When cultural and religious ideas rationalize and sanction inequalities, such as the control of women's bodies, she argues that multiculturalism will cost women their rights (Okin 1999; Cohen et al. 1999). In other words, she highlights the possibility of multiculturalism masking romanticism and, thereby, threatening women's rights. It is difficult to argue for cultural rights that are oppressive since, as Young argues, unchecked claims to multiculturalism can stifle social justice. The desire to embrace a group's rights should not entail the suppression of individual rights or liberty. In a liberal democratic society, a meaningful individual choice is to be guaranteed (Kymlicka 1991). Group rights should only apply to a social group that itself accepts liberal democratic principles because liberal values require both individual freedom of choice and a secure cultural context from which individuals can make their own choices (Kymlicka 1991, p. 169). Contributing to the same argument, Poulter argues that cultural tolerance should not be a "cloak" for promoting oppressive and unjust practices (Poulter 1986, p. 593).

Cultural relativism as a position has had a significant influence on the discourse about FGC. In April/May of 2010, for instance, the American Academy of Pediatrics (AAP) issued a policy statement regarding changes in the federal ban of all forms of FGC procedure (American Academy of Pediatrics 2010). This nationally recognized medical organization called for pediatrician physicians to be allowed to perform a "ceremonial pinprick or nick" on the clitoris of newborn girls from communities that embrace FGC as a way of expressing sensitivity to the cultural and religious values that motivate parents to seek this procedure for their daughters (Committee on Bioethics 2010, p. 1092; Belluck 2010). The academy defended its position, arguing that "it was a way to potentially forestall what it viewed as the more dire consequence of outright prohibition, of the practice which left parents with no option other than sending their daughters outside the U.S, where they would be subjected to more 'disfiguring and life-threatening procedures in their native countries'" (Committee on Bioethics 2010, p. 1092; American Academy of Pediatrics 2010). Doriane L. Coleman defended the "nick" or "pinprick" form of genital cutting, arguing that the procedure was less severe than the commonly acceptable male circumcision. Hospitals must either perform this procedure on girls if requested or stop circumcising boys, she added (Coleman 1998, pp. 717–83). Needless to say, the AAP's move caused uproar amongst human rights activists and feminists, as theirraised concerns related to the implications of recommending a medically unnecessary surgical procedure on the human rights of children.

Those who defend FGC using cultural relativism as an approach have tended to dismiss issues such as health issues and patriarchal ideals associated with the practice, calling them an exaggeration that ignores cultural values that the practice is embedded in. As Garey Goldberg explains, medical research findings show that a high percentage of cut women can have regular and fulfilling sex lives, with the ability to reach orgasm, contrary to the assumptions of moral universalists (Goldberg 2012). F. A. Ahmadu, an insider, has criticized the use of the Western norm of sexuality to critique FGC, arguing that cultural patterns structure norms and experiences differently. Ahmadu argues that the assumption that women in all societies desire sexual fulfillment is misleading since sexuality is socially defined (Ahmadu 2000, p. 284). Amede Obiora explains how conceptions of human dignity tend to be indeterminate and contingent such that an act one may condemn as a depreciative of human dignity may have been enacted by its practitioners as an enhancement of human dignity, and that which may be construed as cruel and in violation of Article 5 may be embraced in cultures where it is practiced as a "technology of the body" (Obiora 1997, p. 277). Shweder adds to this argument by highlighting the need for recognizing the functional role of FGC in communities where it is used to promote social bonding and the socialization of members (Shweider 2000, p. 7). In arguing for group rights and multiculturalism, cultural relativism as an ideology is compelling since

it calls for the recognition of the reality of social differences. The recognition and respect of cultural values, such as those associated with specific practices like FGC, are perceived to be fundamental. This, to cultural relativism, is to claim rights and social justice for the marginalized (Rawls 1971).

## 4. Cross-Cultural Universals and Women's Rights

Calls for cross-cultural universals emerge out of critiques of human rights and cultural relativism. While individual and cultural rights claims are compelling, the problem arises when proponents of either position make dogmatic arguments that make it impossible to protect both individual and group rights. Calls for cross-cultural universals emerge out of theories of intersectionality that seek to harmonize and advocate the need to transcend transfixed social boundaries and binary perceptions of reality to embrace social difference that is characterized by gender, ethnicity, class, and sexual orientation (Collins 1990; Crenshaw 1989, 1991). Leslie McCall considers intersectionality theory as the most important contribution to the women's studies initiative (McCall 2005). Originating in black feminist thought, with scholars such as Kimberle Williams Crenshaw (1989, 1991) and Patricia Hill Collins (1990), it seeks to critique the white solipsism within the feminist discourse. The theory recognizes how social identities intersect. It adds to the cross-cultural universals approach in its emphasis on respecting the diverse cultural values that inform multiculturalism and, hence, the respect of both individual and cultural rights.

Cross-cultural universals are also informed by the social change theory. In her argument about social change, Margaret Archer describes how cultures and social institutions as dynamic phenomena are bound to evolve and change to accommodate new realities (Archer 1988). She explains how humans as social agents are the origins of this change since they respond to cultural systems differently. As social agents, they can reinforce a cultural system or resist its influence when it is perceived to inhibit individual well-being. Even though culture acts on humans once agreed upon, it is important to remember that it is a product of human agency (Archer 1988, pp. 77–78, 143). In essence, therefore, reality undergoes social construction, and this process is often imbued with human error, which makes it vulnerable to further changes. It is for this reason that social change is inevitable. Cultural and institutional change takes place through a process known as cultural appraisal. As Archer explained, this is a social review process through which individuals and their communities assess and reflect upon what is dysfunctional, irrelevant, implausible, unconvincing, and morally unacceptable to determine what cultural values maintain relevance in society. Daniel A. Gordon, C. and Frances. Goldscheider, and Mohammed. J. Abbasi-Shavazi argued that one characteristic of social change is social assimilation, a social process through which generational transition occurs and leads to distinct ethnocultural characteristics fading in strength over time (Gordon 1991; Goldscheider and Goldscheider 1987; Abbasi-Shavazi and McDonald 2000). Ethno-cultural values can either be abandoned or reinforced depending on special characteristics, such as education, skills, culture, opportunities, and social discrimination. The philosopher Alasdair MacIntyre described discarded and abandoned values as "dead tradition" and those that are reinforced and retained as "living" tradition (MacIntyre 2007, p. 222). Those who argue for cross-cultural universals see this process of social change as an inevitable fact of social reality.

Discussions that have examined the conflict between moral universalism and cultural relativism draw from the attempts to protect both individual and group freedoms. As argued by Robert D. Sloane, attempts to reconcile both rights' claims have generated fundamental questions: (1) What criteria are to be used to determine whether "cultural values" violate "universal" standards? (2) Can group and human rights be reconciled toward the advancement of human rights in a world that is embracing multiculturalism? (3) How can basic standards of social justice be ensured and upheld as individual and group rights are protected? (4) Most importantly, how does the conflict between collective and individual rights impact efforts to promote women's rights? (Sloane 2001, p. 531). While some have argued that the controversy between moral universalism and cultural

relativism has no common ground, scholars such as Bret Billet and Sloane envisioned a common ground in cross-cultural universals. Sloane and Billet argued for a middle ground that draws from identified general guidelines. These guidelines should be preceded by an acknowledgment of the fact that all cultures, both Western and non-Western, suffer from ethnocentrism and a commitment to preserving the values, beliefs, and morals that define their people (Sloane 2001, p. 580; Billet 2007, pp. 183–85). They have argued for the possibility of a strategic reconciliation between universal and cultural values (Sloane 2001, p. 580; Billet 2007, pp. 183–85). In essence, they contest the claim that women's rights are under threat in a multicultural world.

Sloane (2001), in particular, argues that the cross-cultural approach should begin with the presumption that universal human rights represent the desirable end-state to moral universalists. It involves inquiry of how to establish effective conditions under which international human rights can receive respect in a global order that is characterized by cultural pluralism. The answer does not lie in the manipulation and redeployment of each culture's internal resources in the service of human rights, she argued. It is the objective of cross-cultural universals to tap into "shareable mores" each society provides to arrive at a consensus based on basic overlapping values that most cultures are bound to respect (Sloane 2001, p. 580).

Billet explains how the negotiation of the seemingly unbridgeable chasm between moral universalism and cultural relativism is part of the socio-cultural process of cultural appraisal that any given community must undergo (Billet 2007, p. 184). Attempts at exploring a middle ground to promote a cross-cultural universal are respectful of both cultural and human rights positions. Elisabeth Reichert adopts a similar position in recognizing the importance of reflecting upon significant questions that arise out of multiculturalism as a discourse. It is important to ask questions such as: Whose voices are being heard in a specific culture? Who defines culture and who has the power to define it? Who benefits from the said definition and who does not? How can all voices be heard? How can education be designed to empower those without power? (Reichert 2006, p. 29). A deep understanding of diverse cultures is necessary to promote cross-cultural universals, argues E. Reichert (2006). Ultimately, voices for or against practices such as female genital cutting must should emerge out of cultural appraisal process and amongst the women themselves within the communities that embrace such practices.

## 5. Conclusions

This article explored two moral positions informing the discourse on human rights and group rights, namely, moral universalism and cultural relativism. Scholarly literature on the evolution of these positions was examined as they relate to the discourse on women's rights and, specifically, as they relate to African communities that embrace female genital cutting. Most importantly, the discourse explored the implications of this moral position for womens' rights and African communities that embrace practices, such as female genital cutting. The most controversial aspect of these positions has been when the negotiation between human rights and groups' rights seems impossible. The article demonstrates how compelling the arguments on both sides are and how the multicultural argument complicates efforts toward promoting women's rights. As Okin observed, the central question is whether it is possible to achieve multiculturalism that treats all persons like each other's moral equals when it tends to resist reconciliation with egalitarian convictions. The article demonstrates further how the seemingly irreconcilable moral positions can be reconciled using a cross-cultural universals approach through the cultural appraisal process that naturally occurs with social change.

While acknowledging the implications of multiculturalism and the human rights discourse on the practice of female genital cutting, the article describes the dilemma that is induced by the seemingly irreconcilable arguments for human rights versus group rights and how conflicted those who embrace the procedure find themselves. It is, however, demonstrated by this discussion that calls for multiculturalism and cultural diversity are

healthy and vital for the invigoration of lost cultural heritage. Each position has moral implications for women's rights and whether female genital cutting ought to be condemned and eradicated in efforts to protect the rights of those exposed to them or whether the practice should be left alone or retained in respect for the cultural rights of the communities that embrace them should be a matter of cultural appraisal and an embrace of cultural universalism if a community chooses to. This article demonstrates how efforts to resolve the ensuing dilemma through cultural universals is an attempt at respecting arguments in favor of both positions. The discourse on intersectionality and social change theories present this possibility by arguing for the recognition of diversity as a social reality of the modern pluralistic and globalizing world.

While the recognition and embrace of difference and human rights are essential, as argued above, extreme advocacy of either position influences and may sometimes undermine efforts to protect the rights of either group. The embrace of cross-cultural universals is the most palatable approach to ensuring that reconciliation efforts are made to respect both human and cultural rights. This means, as coerced uninformed FGC practices that involve children arediscouraged, condemned, and eradicated, consented informed uncoerced practices should be respected as individual rights that involve freedom of choice. Ultimately, the argument leans toward discouraging the use of culture or group rights as a cloak for undermining human rights.

**Funding:** This research received no external funding.

**Institutional Review Board Statement:** IRB process was waived for this student as human subjects were not involved.

**Informed Consent Statement:** Not applicable.

**Data Availability Statement:** Not applicable.

**Conflicts of Interest:** The authors declare no conflict of interest.

## Note

1     The author recognizes the fact that there are other terminologies, such as female circumcision and female genital mutilation, that are in use to describe this practice that imply insensitive connotations. I choose to use the term female genital cutting to avoid misconceptions associated with other terms.

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
