# Peer review of "Multiculturalism and Women’s Rights: Implications for Victims of Female Genital Cutting"

_religions, doi:10.3390/rel13040367_

Round 1

Reviewer 1 Report

This article makes a solid contribution in understanding the issues related to the controversial practice of female genital cutting (FGC). The author has convincingly presented both sides of the arguments from women's rights and cultural rights. I was intrigued by theoretically sound arguments to reconcile the issues from cross-cultural universalism and intersectionality emerging as a bridge between the two opposing positions.

There are some awkward constructions at # 95-98 and #299-301. Please read the sentences carefully and refine them to make sense to the readers. I was a bit confused by the author's narration of "R.D. Sloane" at # 333 and then cite the same scholar as "Sloan 2001" in parenthesis. Is "Sloane" and "Sloan" one and the same person? Which is correct? Please clarify this at #s 339, 341 and 350. The author must be careful here.

On the whole, the author presents a balanced perspective on the issue of FGC. Ultimately, the protest must emerge among the women themselves within the Muslim community against this practice. Of course it is done to the children and when they grow up they cannot do anything about it.   

Author Response

I have addressed the comments you suggested. 

Reviewer 2 Report

I have attached my comments as a word document

Author Response

I have responded to comments made and here is the clean copy.
